# Characterization of acquired β-lactamases in *Pseudomonas aeruginosa* and quantification of their contributions to resistance

Karl A. Glen,[1] Iain L. Lamont[1]

**ABSTRACT**  *Pseudomonas aeruginosa* is a highly problematic opportunistic pathogen that causes a range of different infections. Infections are commonly treated with β-lactam antibiotics, including cephalosporins, monobactams, penicillins, and carbapenems, with carbapenems regarded as antibiotics of last resort. Isolates of *P. aeruginosa* can contain horizontally acquired *bla* genes encoding β-lactamase enzymes, but the extent to which these contribute to β-lactam resistance in this species has not been systematically quantified. The overall aim of this research was to address this knowledge gap by quantifying the frequency of β-lactamase-encoding genes in *P. aeruginosa* and by determining the effects of β-lactamases on susceptibility of *P. aeruginosa* to β-lactams. Genome analysis showed that β-lactamase-encoding genes are present in 3% of *P. aeruginosa* but are enriched in carbapenem-resistant isolates (35%). To determine the substrate antibiotics, 10 β-lactamases were expressed from an integrative plasmid in the chromosome of *P. aeruginosa* reference strain PAO1. The β-lactamases reduced susceptibility to a variety of clinically used antibiotics, including carbapenems (meropenem, imipenem), penicillins (ticarcillin, piperacillin), cephalosporins (ceftazidime, cefepime), and a monobactam (aztreonam). Different enzymes acted on different β-lactams. β-lactamases encoded by the genomes of *P. aeruginosa* clinical isolates had similar effects to the enzymes expressed in strain PAO1. Genome engineering was used to delete β-lactamase-encoding genes from three carbapenem-resistant clinical isolates and increased susceptibility to substrate β-lactams. Our findings demonstrate that acquired β-lactamases play an important role in β-lactam resistance in *P. aeruginosa*, identifying substrate antibiotics for a range of enzymes and quantifying their contributions to resistance.

**IMPORTANCE**  *Pseudomonas aeruginosa* is an extremely problematic pathogen, with isolates that are resistant to the carbapenem class of β-lactam antibiotics being in critical need of new therapies. Genes encoding β-lactamase enzymes that degrade β-lactam antibiotics can be present in *P. aeruginosa*, including carbapenem-resistant isolates. Here, we show that β-lactamase genes are over-represented in carbapenem-resistant isolates, indicating their key role in resistance. We also show that different β-lactamases alter susceptibility of *P. aeruginosa* to different β-lactam antibiotics and quantify the effects of selected enzymes on β-lactam susceptibility. This research significantly advances the understanding of the contributions of acquired β-lactamases to antibiotic resistance, including carbapenem resistance, in *P. aeruginosa* and by implication in other species. It has potential to expedite development of methods that use whole genome sequencing of infecting bacteria to inform antibiotic treatment, allowing more effective use of antibiotics, and facilitate the development of new antibiotics.

Address correspondence to Iain L. Lamont, iain.lamont@otago.ac.nz.

The authors declare no conflict of interest.

See the funding table on p. 12.

**KEYWORDS** *Pseudomonas aeruginosa*, beta-lactamases, carbapenems, carbapenemase, cephalosporins, monobactams, penicillins, horizontal gene transfer, tazobactam, mobile genetic elements

The opportunistic pathogen *Pseudomonas aeruginosa* can cause severe and life-threatening infections in burn wounds, surgical sites, lungs of people with cystic fibrosis, and immunocompromised patients (1–7). β-lactam antibiotics play a key role in the treatment of *P. aeruginosa* infections (8, 9), with a large variety of β-lactams being used, including the penicillins ticarcillin and piperacillin, the cephalosporins ceftazidime and cefepime, the carbapenems imipenem and meropenem, and the monobactam aztreonam (3, 9, 10). Of particular importance is the carbapenem subclass as they are generally reserved as a last line of defense in the treatment of known or suspected multidrug-resistant bacteria due to their ability to resist hydrolysis by most β-lactamases (11–13). However, treatment is becoming more difficult because of the increasing frequency of β-lactam-resistant isolates (2, 14, 15). Indeed, the World Health Organization has classified carbapenem-resistant *P. aeruginosa* as a high-priority group for the development of new treatments (16). Novel β-lactams that inhibit the growth of *P. aeruginosa* have been developed, but resistance to these can arise in the same way as resistance to currently used antibiotics (17, 18).

There are many mechanisms by which β-lactam-susceptible strains of *P. aeruginosa* can become resistant. These include mutations that increase expression or alter substrate specificity of multidrug efflux pumps and the intrinsic low-activity β-lactamases AmpC and an OXA-50-like β-lactamase, resulting in increased export and degradation of β-lactams (19–23); mutations in porins responsible for uptake of β-lactams (24–26); mutations in penicillin-binding proteins that are the targets of β-lactam antibiotics; and acquisition of β-lactamases through horizontal gene transfer (27–32). These can be combinatorial, with resistant isolates often having several different mechanisms for reducing susceptibility to β-lactams.

β-lactamases hydrolyze the β-lactam ring of substrate compounds, inactivating the antibiotics (33). These enzymes are categorized into four classes (A to D) based on sequence homology. Enzymes in classes A, C, and D have a catalytic serine for substrate hydrolysis, whereas class B are metallo-enzymes that catalyze the hydrolysis of β-lactam rings using a metal ion, most commonly a zinc ion (34, 35). Collectively, serine β-lactamases can hydrolyze all β-lactams in clinical use, and metallo-β-lactamases can hydrolyze all clinically used β-lactams except monobactams, although most individual enzymes are only active against a subset of compounds (32, 36, 37). β-lactamases capable of degrading carbapenems (carbapenemases) are especially problematic due to the last-resort role of carbapenems in managing severe infections by multidrug-resistant *P. aeruginosa*. Carbapenemases can be present in carbapenem-resistant isolates of *P. aeruginosa* that are often a cause of severe infections (27, 38–41). β-lactamase inhibitors, such as tazobactam, clavulanate, avibactam, vaborbactam, and relebactam, that inhibit some serine β-lactamases can enhance the activity of β-lactam antibiotics by reducing the rate of antibiotic hydrolysis and are often co-administered with β-lactams (42–47).

*P. aeruginosa* isolates can acquire *bla* genes that encode β-lactamases through horizontal gene transfer (28–32). The *bla* genes are typically carried on mobile elements, predominantly chromosomally located composite transposons, although they are on plasmids in some cases (48, 49). Horizontal gene transfer can occur between different species of bacteria, providing the potential for *P. aeruginosa* to acquire β-lactamases from unrelated species, such as from a variety of *Enterobacteriaceae* species (50).

Although the substrate antibiotics of some β-lactamases have been determined (31, 51, 52), new enzyme variants are regularly discovered (53–55), and the substrates of most of these are unknown. Consequently, in many cases, it is not known which antibiotics will be effective against a β-lactamase-containing isolate. Furthermore, although many studies have reported the presence of horizontally acquired β-lactamases in *P. aeruginosa*, there is currently very limited quantitative information on the extent to

which these enzymes reduce β-lactam susceptibility in this species. The overall goal of this research was to quantify the effects of acquired β-lactamases on resistance of *P. aeruginosa* to different β-lactams.

## RESULTS

### Prevalence of acquired β-lactamases in *P. aeruginosa*

Determining the frequency of acquired *bla* genes within the *P. aeruginosa* population is a first step towards quantifying their contribution to resistance in this species. The genomes of 858 *P. aeruginosa* isolates with unknown antibiotic resistance phenotypes from the International *Pseudomonas* Consortium Database (56, 57) were interrogated for the presence of *bla* genes. The isolates (Table S1) were of worldwide origin (five continents, multiple different countries and centers) and were from a range of different infections and from environmental and veterinary sources. Of the 858 isolates, 26 (3%) had acquired *bla* genes. These isolates harbored a wide range of acquired genes (Table 1; Table S1). Genes encoding class D enzymes were the most frequent, with OXA-10 being the most prevalent of these (39% of class D enzymes). Class A enzymes were the next most common, with CARB-2 being the most prevalent of these (57%). Class B enzymes were the least frequent, with VIM-2 being the most prevalent (50%). No acquired Class C β-lactamases were found.

We next investigated whether *bla* genes are enriched in isolates of *P. aeruginosa* that are resistant to meropenem, a carbapenem antibiotic. The frequency of acquired β-lactamases was determined in 238 independent meropenem-resistant *P. aeruginosa* isolates in our collection (clinical isolates of worldwide origin) and from a previous study (clinical isolates from hospitals in Europe) (58) (Table 1; Table S2). Sixty-one of these isolates (35%) had acquired one *bla* gene and 23 had acquired multiple genes (Table S2), indicating a strong selection for β-lactamases in carbapenem-resistant isolates. The most common β-lactamases found in carbapenem-resistant isolates for each class were GES-5 for class A, VIM-2 for class B, and OXA-2 for class D, all of which have activity against carbapenems, including meropenem. Meropenem resistance is multi-factorial, and other factors, such as absence of OprD or increased expression of efflux pumps, may also contribute to the meropenem-resistant phenotype of these isolates.

The presence of different *bla* genes in different isolates implies multiple independent acquisitions of *bla* genes into *P. aeruginosa*, although the presence of identical

**TABLE 1** β-lactamases in uncharacterized and meropenem-resistant *P. aeruginosa* isolates

| Class A | Number of genomes | Class B | Number of genomes | Class D | Number of genomes |
|---|---|---|---|---|---|
| β-lactamases encoded by 858 uncharacterized *P. aeruginosa* genomes | | | | | |
| CARB-2 | 8 | IMP-14 | 1 | OXA-2 | 1 |
| VEB-1 | 4 | IMP-54 | 1 | OXA-4 | 1 |
| VEB-2 | 3 | SPM-1 | 2 | OXA-10 | 8 |
| | | VIM-2 | 4 | OXA-21 | 1 |
| | | | | OXA-56 | 6 |
| | | | | OXA-129 | 1 |
| β-lactamases encoded by genomes of 238 meropenem-resistant *P. aeruginosa* | | | | | |
| CARB-2 | 4 | IMP-1 | 1 | LCR-1 | 2 |
| GES-1 | 5 | IMP-7 | 1 | OXA-1 | 1 |
| GES-5 | 6 | IMP-31 | 2 | OXA-2 | 13 |
| GES-9 | 1 | VIM-2 | 18 | OXA-4 | 3 |
| GES-19 | 4 | VIM-4 | 3 | OXA-10 | 8 |
| GES-20 | 3 | VIM-47 | 4 | OXA-17 | 1 |
| KPC-2 | 1 | | | OXA-35 | 2 |
| PER-1 | 1 | | | OXA-36 | 1 |
| SHV-12 | 2 | | | OXA-74 | 1 |
| VEB-1 | 4 | | | OXA-233 | 1 |

*bla* genes in some cases suggested the possible occurrence of multiple independent isolates of a clonal strain. A phylogenetic tree was therefore constructed with genomes of 64 representative *P. aeruginosa* isolates from five different continents and including 20 carbapenem-resistant isolates, 21 susceptible isolates, and 23 isolates of unknown phenotype (Fig. 1). An extended phylogenetic tree with all isolates containing acquired β-lactamase was also constructed (Fig. S1).

The genomes (Fig. 1; Fig. S1) have a similar distribution to a larger tree that represents the breadth of the *P. aeruginosa* species (60), showing that it encompasses most of the genetic diversity of the *P. aeruginosa* species, except for the absence of isolates related to the PA7 subgroup. Acquired β-lactamases were present in a wide distribution of carbapenem-resistant isolates and isolates of unknown phenotype (Fig. 1; Fig. S1). The

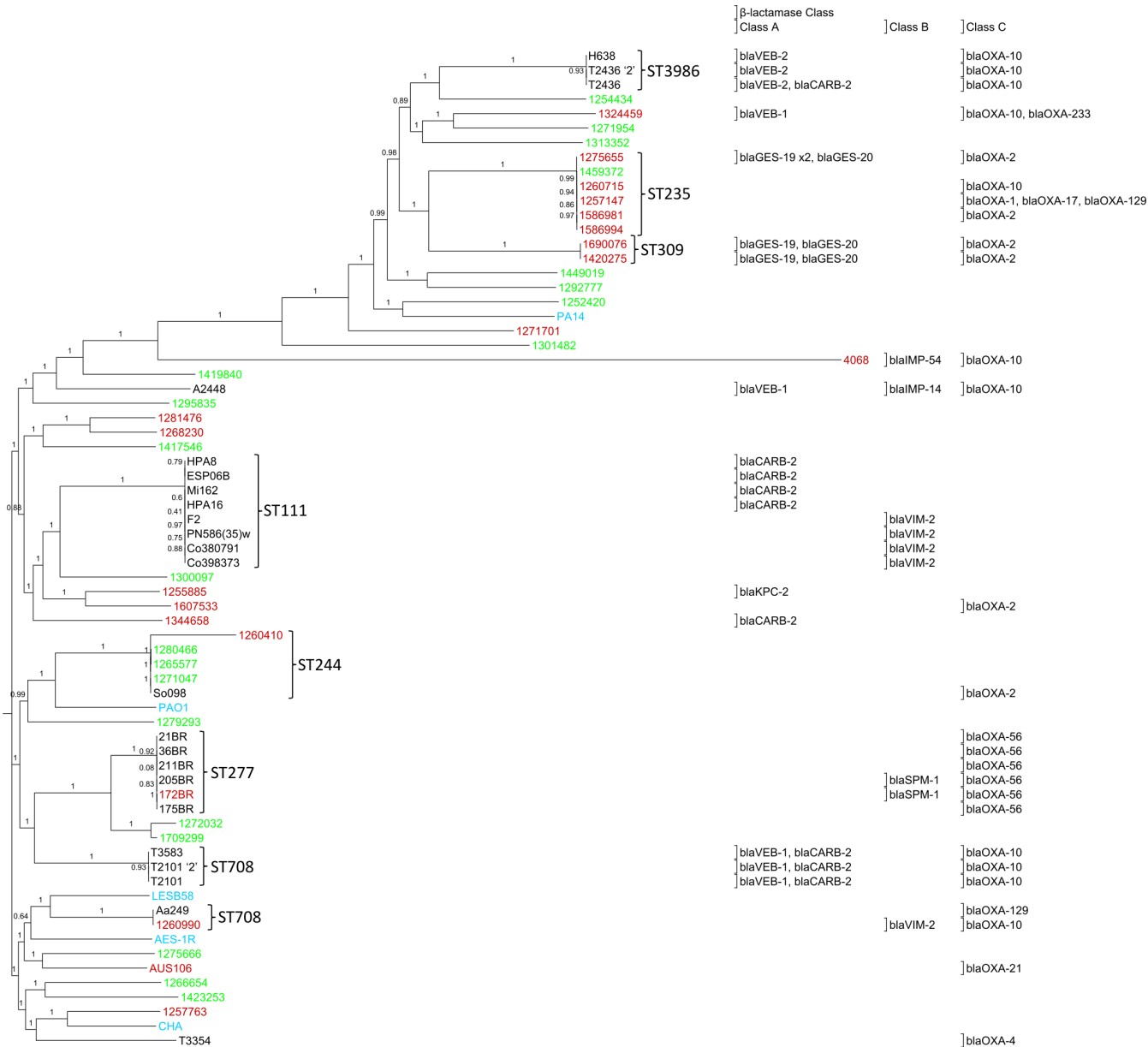

**FIG 1** Phylogenetic tree of representative β-lactamase containing *P. aeruginosa* isolates. The tree comprises 37 isolates with acquired β-lactamases and 27 isolates with no acquired β-lactamases (Table S3). The identities of the β-lactamases are shown. Meropenem-resistant isolates are shown in red, meropenem-susceptible isolates in green, and isolates with unknown phenotype in black. Five *P. aeruginosa* reference strains (in blue) are also included (59). Sequence types (ST) are shown.

*P. aeruginosa* isolates have a few groups of closely related strains harboring the same β-lactamases (Fig. 1; Fig. S1), indicating that these clones are highly successful. Notably, there are multiple isolates of the highly virulent clone ST235 (61–63), and these isolates have acquired a range of β-lactamases. There are also multiple isolates of other high-risk clones, including ST111 (61, 63) and ST277 (61, 64–66).

## Substrate specificities of β-lactamases

Substrate antibiotics were determined for a range of β-lactamases to better understand the contributions of different enzymes to resistance. Genes encoding β-lactamases from *P. aeruginosa* isolates were cloned and expressed in *P. aeruginosa* PAO1, and the effects on MICs of representative antibiotics from all β-lactam classes were measured (Table 2). The β-lactamases chosen that were prevalent in clinical isolates (VIM-2, OXA-10) (Table 1) had not been investigated previously (OXA-21, OXA-129, and the OXA-2-related OXA-226) and/or in β-lactamase families associated with carbapenem resistance (KPC-2, GES-19, GES-20, VIM-2, SPM-1. and IMP-54) (13, 42, 67). Most *bla* genes are present in the chromosome or on a low copy number of plasmids (27), and so a chromosome-integrating vector (pSW196) was used to represent the normal situation. MIC testing was performed on *P. aeruginosa* PAO1 expressing the β-lactamases (Table 2).

The 10 β-lactamases all increased the MIC for antibiotics in at least one class of β-lactams, with the spectrum of substrates and the extent of the increase differing between enzymes. The three class A β-lactamases (KPC-2, GES-19, and GES-20) and three class B enzymes (VIM-2, SPM-1, and IMP-54) all increased the MICs for at least six of eight β-lactams tested, with KPC-2 increasing the MIC of all β-lactams. Class B β-lactamases did not increase aztreonam MIC, but some enzymes in classes A and D did. Classes A and B enzymes had carbapenemase activity, increasing the MIC for at least one carbapenem, whereas the four class D β-lactamases had little or no carbapenemase activity.

## MICs for β-lactamase-containing isolates

The MICs of *P. aeruginosa* containing a variety of acquired *bla* genes were determined for comparison with strain PAO1 expressing the same genes (Table 3).

Most *bla*-containing isolates had higher MICs than reference strain *P. aeruginosa* PAO1 for all β-lactams tested (Tables 2 and 3). However, the results were consistent with the

**TABLE 2**  MICs of *P. aeruginosa* PAO1 expressing *bla* genes[c]

| Expressed β-lactamase | Antibiotic MICs mg/L[a] | | | | | | | |
| --- | --- | --- | --- | --- | --- | --- | --- | --- |
| | Carbapenems | | Penicillins | | | Cephalosporins | | Monobactam |
| | MEM | IMI | CARB | PIP | TIC | CTZ | CPM | AZT |
| None[b] | 0.5 | 2 | 16 | 2 | 8 | 0.5 | 0.5 | 2 |
| Empty vector | 0.5 | 2 | 16 | 2 | 8 | 0.5 | 0.5 | 2 |
| Class A | | | | | | | | |
| KPC-2 | 8 | 4 | 512 | 16 | 128 | 4 | 16 | 32 |
| GES-19 | 2 | 2 | 512 | 16 | 256 | 512 | 64 | 64 |
| GES-20 | 16 | 2 | 128 | 16 | 32 | 8 | 2 | 2 |
| Class B | | | | | | | | |
| VIM-2 | 16 | 16 | >1024 | 16 | 512 | 16 | 8 | 2 |
| SPM-1 | 32 | 8 | 128 | 8 | 32 | 256 | 128 | 2 |
| IMP-54 | 32 | 4 | 32 | 2 | 16 | 64 | 64 | 2 |
| Class D | | | | | | | | |
| OXA-226 | 0.5 | 2 | 32 | 2 | 32 | 16 | 1 | 8 |
| OXA-10 | 0.5 | 1 | 128 | 8 | 64 | 0.5 | 2 | 4 |
| OXA-129 | 0.5 | 1 | 32 | 4 | 32 | 0.5 | 1 | 2 |
| OXA-21 | 1 | 1 | 256 | 8 | 128 | 2 | 1 | 2 |

[a]Median of three replicates, with all replicate values in Table S4.
[b]None, strain PAO1; empty vector, PAO1 containing pSW196 vector with no cloned *bla* gene.
[c]MEM = Meropenem, IMI = Imipenem, PIP = Piperacillin, TIC = Ticarcillin, CTZ = ceftazidime, CPM = cefepime, AZT =Aztreonam, CARB = Carbenicillin.

**TABLE 3**  MICs of isolates with acquired β-lactamases

| Isolate | Acquired β-lactamase | Antibiotic MICs mg/L[a] | | | | | | | |
|---|---|---|---|---|---|---|---|---|---|
| | | Carbapenems | | Penicillins | | | Cephalosporins | | Monobactam |
| | | MEM | IMI | CARB | PIP | TIC | CTZ | CPM | AZT |
| PA01 | None | 0.5 | 2 | 16 | 2 | 8 | 0.5 | 0.5 | 2 |
| Class A | | | | | | | | | |
| 1255885 | KPC-2 | **1024** | **256** | >2048 | **256** | **2048** | **64** | **256** | **512** |
| Class D | | | | | | | | | |
| 1257147 | OXA-1/17/129 | **8** | 1 | 1024 | **128** | **512** | 4 | **64** | 16 |
| 1260715 | OXA-10 | **16** | **8** | 256 | 16 | **128** | 1 | 2 | 2 |
| 1586981 | OXA-2 | **32** | **16** | 1024 | **128** | **256** | **64** | 16 | **32** |
| 1607533 | OXA-2 | **8** | **16** | 128 | 16 | 32 | 8 | 8 | 8 |
| AUS106 | OXA-21 | **16** | **8** | 512 | 64 | **256** | 8 | 8 | 8 |
| Classes A and D | | | | | | | | | |
| 1275655 | OXA-2, GES-19/19/20 | **128** | **32** | >2048 | 128 | **1024** | 2048 | **256** | **256** |
| 1324459 | OXA-10/233 VEB-1 | **64** | **16** | 2048 | 32 | **1024** | 1024 | **256** | **1024** |
| 1420275 | OXA-2, GES-19/20 | **256** | **34** | 512 | 128 | **256** | 2048 | **256** | **128** |
| 1690076 | OXA-2, GES-19/20 | **16** | **8** | 1024 | 32 | **256** | 256 | 64 | **32** |
| Classes B and D | | | | | | | | | |
| 1260990 | OXA-10, VIM-2 | **128** | **128** | >2048 | 64 | **2048** | 16 | 16 | 8 |
| 172BR | SPM-1, OXA-56 | **256** | **128** | 1024 | 32 | **128** | 256 | **128** | 16 |
| 4068 | OXA-10, IMP-54 | **>2048** | **256** | >2048 | **128** | **>2048** | 512 | **2048** | **128** |

[a]Median of three replicates, with all replicate values shown in Table S5. Values at or above the threshold for resistance (68) are shown in bold font.

effects of cloned *bla* genes expressed in strain PAO1. All *P. aeruginosa* strains with a β-lactamase that increased carbapenem MIC in strain PAO1 (Table 2) were resistant to both carbapenems meropenem and imipenem (Table 3). Class B and D enzymes had little or no effect on the MIC for aztreonam in strain PAO1, and four isolates that were susceptible to aztreonam had only class B and/or D β-lactamases. All the clinical isolates that were susceptible to cephalosporins had only class D β-lactamases, which showed little to no activity against cephalosporins in strain PAO1 (Table 2).

## Deletion of carbapenemase genes from clinical isolates

In order to quantify the contributions of carbapenemases to resistance, genes encoding three carbapenemases VIM-2, GES-20 and IMP-54 were deleted from the genomes of *P. aeruginosa* isolates. These genes were selected for deletion because they conferred the highest increases in meropenem MIC when expressed in *P. aeruginosa* PAO1 (Table 2), because carbapenems are "last-resort" antibiotics in treatment of *P. aeruginosa* infections, and because complete genome sequences necessary for engineering the deletions were available. The genes encoding GES-19 and GES-20 are adjacent to each other and flanked by repetitive DNA sequences, making it necessary for technical reasons to delete *blaGES19* as well as *blaGES20*. MIC testing was performed on the engineered deletion-containing mutants (Table 4).

Deletion of the *blaVIM-2* gene from the genome of isolate 1260990 reduced the MIC for all antibiotics except aztreonam, consistent with the effect of this gene on strain PAO1 (Table 2), although the bacteria remained resistant to carbapenems and ticarcillin. Deletion of the *blaGES-19* and *blaGES-20* genes from the genome of isolate 1420275 reduced the MIC of all antibiotics tested, making the bacteria susceptible to penicillins, cephalosporins, and aztreonam and consistent with the activity of class A enzymes against all classes of β-lactams. Deletion of the *blaIMP-54* gene from isolate 172BR also reduced the MICs of all β-lactams although only rendered the isolate susceptible to ceftazidime. Overall, none of the gene deletions were sufficient to render the isolates carbapenem-susceptible, but all three deletions reduced carbapenem MIC. The decreases in MICs caused by deleting *bla* genes are consistent with the increases in

**TABLE 4** MICs of β-lactams for *P. aeruginosa* isolates with deletions of acquired carbapenemase genes

| Isolate | Antibiotic MICs mg/L[a] | | | | | | | |
| --- | --- | --- | --- | --- | --- | --- | --- | --- |
| | Carbapenems | | Penicillins | | | Cephalosporins | | Monobactam |
| | MEM | IMI | CARB | PIP | TIC | CTZ | CPM | AZT |
| PA01 | 0.5 | 2 | 16 | 2 | 8 | 0.5 | 0.5 | 2 |
| 1260990 | **128** | **128** | 2048 | 64 | **1024** | 16 | 16 | 8 |
| 1260990 ΔVIM-2 | **16** | **16** | 512 | 32 | **256** | 2 | 8 | 8 |
| 1420275 | **256** | **32** | >2048 | **128** | 1024 | 2048 | 256 | 128 |
| 1420275 ΔGES-19/20 | **16** | **8** | 256 | 8 | 64 | 4 | 8 | 32 |
| 4068 | **>2048** | **256** | >2048 | **128** | **>2048** | 512 | 2048 | 128 |
| 4068ΔIMP-54 | **256** | **8** | >2048 | **128** | 2048 | 4 | **64** | **64** |

[a]Median of three replicates, all replicates are in Table S6. Values at or above the threshold for resistance (68) are shown in bold font.

MICs that the β-lactamases provide when expressed in *P. aeruginosa* PAO1 (Table 2). All three of these isolates had deletions or other mutations that inactivate OprD, the porin that is responsible for uptake of carbapenems, and the absence of functional OprD may explain their resistance to carbapenems despite the absence of the *bla* genes (69, 70).

## Effectiveness of tazobactam on *P. aeruginosa* containing acquired β-lactamase genes

The β-lactamase inhibitor tazobactam is commonly co-administered with β-lactams to restore their function against β-lactamase-containing isolates (71, 72). To investigate the effectiveness of tazobactam against *P. aeruginosa* containing acquired β-lactamases, antibiotic MICs were determined for a *P. aeruginosa* PAO1 mutant that has a deletion of *ampC*. This mutant was used to avoid any confounding effects of the intrinsic β-lactamase AmpC, as inhibition of AmpC by tazobactam might affect the MICs. β-lactams that had a higher MIC in the presence of one or more β-lactamases (Table 2) were tested (Table 5).

Tazobactam had activity against two class D β-lactamases, OXA-21 and to a lesser extent OXA-226. Class A enzymes (KPC-2 and GES19/20) were not inhibited by tazobactam, consistent with previous findings for KPC-2 (47, 73). Class B β-lactamases (metallo-β-lactamase, VIM-2, IMP-54, and SPM-1) were also not inhibited by tazobactam, which is known to lack activity against class B enzymes (33).

## *bla* genes are parts of complex mobile genetic elements

Mobile genetic elements allow for the spread of β-lactamase-encoding genes and a large range of other antibiotic-resistance genes (27, 74). The mobile elements that contained β-lactamases were investigated for carbapenem-resistant isolates for which complete (fully assembled) genome sequences were available to determine how the β-lactamases were likely acquired and spread. Genes encoding VIM-2 and OXA-10 (Fig. S2a), and GES-19, GES-20, and OXA-2 (Fig. S2b) are on transposons in the chromosomes of the isolates 1260990 and 1420275, respectively. The *blaKPC-2* gene present in isolate

**TABLE 5** Effect of tazobactam on *P. aeruginosa* PAO1 *ampC* expressing acquired β-lactamases

| β-lactamase | MEM | MEM+TZ[b] | β-lactamase | CPM | CPM + TZ | β-lactamase | TIC | TIC+TZ | β-lactamase | CTZ | CTZ+TZ |
| --- | --- | --- | --- | --- | --- | --- | --- | --- | --- | --- | --- |
| None[c] | 0.5 | 0.5 | None | 1 | 1 | NA | 8 | 8 | None | 0.5 | 0.5 |
| GES-20 | 16 | 16 | KPC-2 | 16 | 16 | OXA-10 | 64 | 64 | OXA-226 | **16** | **8** |
| VIM-2 | 16 | 16 | GES-19 | 64 | 64 | OXA-129 | 32 | 32 | | | |
| | | | SPM-1 | 128 | 128 | OXA-21 | **128** | **16** | | | |
| | | | IMP-54 | 64 | 64 | OXA-226 | **32** | **16** | | | |

[a]Median of three replicates, all replicates are in Table S7. MICs that were lower in the presence of tazobactam are indicated in bold font.
[b]TZ = Tazobactam.
[c]None, PAO1Δ*ampC*.

**TABLE 6** Effect of plasmid pPAE1255885 on the MICs of *P. aeruginosa* PAO1

| Strain | Antibiotic MICs mg/L[a] | | | | | | | |
|---|---|---|---|---|---|---|---|---|
| | Carbapenems | | Penicillins | | | Cephalosporins | | Monobactam |
| | MEM | IMI | CARB | PIP | TIC | CTZ | CPM | AZT |
| PAO1 | 0.5 | 2 | 16 | 2 | 8 | 0.5 | 0.5 | 2 |
| PAO1 +KPC-2 | 8 | 4 | 512 | 16 | 128 | 4 | 16 | 32 |
| PAO1 +pPAE1255885 | 256 | 128 | >2048 | 256 | 2048 | 32 | 128 | 512 |
| 1255885 | 1024 | 256 | >2048 | 256 | 2048 | 64 | 256 | 512 |

[a]Median of three replicates, all replicate values are in Table S8.

1255885 is encoded by a large plasmid (44 kb), named in this study as pPAE1255885 (Fig. S2c).

The transposon-located resistance genes were predominantly found in non-*Pseudomonas* species, in particular members of the *Enterobacteriaceae*. These included members of the *Aeromonas, Citrobacter, Enterobacter, Klebsiella,* and *Raoultella* genera, indicating that these transposons or the genes on them have likely been acquired by *P. aeruginosa* from other species. The transposons contained multiple genes for resistance to other compounds, notably aminoglycoside antibiotics and also mercury and sulfonamide (Fig. S2).

The *bla*KPC-2 gene is part of a plasmid pPAE1255885 (Fig. S2c) that is very similar to plasmids pPA2047 and pPae-13 (75, 76), as well as plasmids pPA-1 and pPA-2 (BioSamples; SAMN28550649 and SAMN28550875, respectively). The main differences are multiple rearrangement/duplication events, although pPae-13 lacks a transposon present in the other plasmids. Plasmid pPAE1255885 was transformed into *P. aeruginosa* PAO1, and MIC testing was carried out (Table 6). The MIC for *P. aeruginosa* PAO1 containing the KPC-2-encoding pPAE1255885 was significantly higher for all antibiotics tested than for *P. aeruginosa* expressing KPC-2 from the chromosomally integrated vector. The difference in MIC may be due to an uncharacterized antibiotic resistance mechanism on pPAE1255885, or to the multi-copy nature of pPAE1255885, compared with the chromosomally expressed KPC-2 gene. Isolate 1255885 that contains pPAE1255885 had similar MIC values to strain PAO1 harboring the same plasmid (Table 6). This indicates that the majority of the β-lactam resistance in 1255885 is due to the plasmid.

## DISCUSSION

The overall aim of this research was to quantify the contribution of acquired β-lactamases to the resistance of *P. aeruginosa* to β-lactam antibiotics, with a particular focus on carbapenems and carbapenemases. Our findings show that acquired *bla* genes are enriched in carbapenem-resistant isolates. Classes A and B enzymes, although not class D, had carbapenemase activity in *P. aeruginosa*. Deletion of *bla* genes from clinical isolates of *P. aeruginosa* increased susceptibility to multiple β-lactams, including carbapenems and allowed quantification of the effects of the corresponding β-lactamases on MIC. Tazobactam had limited activity against the enzymes characterized in this study.

Quantification of the effects of acquired *bla* genes can contribute to ongoing efforts to predict antibiotic susceptibility of *P. aeruginosa* from genome sequences (77, 78). Knowledge of the substrate specificities of different β-lactamases will also facilitate the application of artificial intelligence for development of new antibiotics and inhibitors (79, 80).

Although the effects of some of the enzymes tested here (KPC-2, GES-19, and VIM-2) had been assessed previously in *E. coli* for a subset of the antibiotics used here (51, 81–83), only SPM-1 and OXA-10 had previously been investigated for their effects in a *P. aeruginosa* reference strain (84, 85). So far, the remaining enzymes GES-20, IMP-54, OXA-21, OXA-129, and OXA-226, had not been characterized for any species. Substrates

of β-lactamases were identified by expressing *bla* genes from an integrative vector that mimics the acquisition of genes into the chromosome of *P. aeruginosa* by integrative mobile genetic elements. This approach enabled an efficient survey of the effects of acquired β-lactamases of different classes on *P. aeruginosa*. Class A β-lactamases collectively increased the MICs of all tested β-lactams, although only KPC-2 increased MICs towards all β-lactams (Table 2). KPC-2 increases the MIC of imipenem, piperacillin, and ceftazidime when expressed in *E. coli* (82, 83), consistent with the results of this study. Although GES-19 and GES-20 are only two amino acid residues different in sequence, they had significantly different effects on β-lactam susceptibility, with GES-19 increasing the MIC of penicillins, cephalosporins, and monobactams, and GES-20, that of meropenem (Table 2). Consistent with these results, deletion of both GES-19 and GES-20 genes from a clinical isolate reduced the MIC of all β-lactams tested (Table 4). Intriguingly, deletion of both GES-19 and GES-20 reduced the MIC for imipenem, whereas expression of the individual genes did not increase the MIC of reference strain PAO1 (Table 2). The effect of co-expression of both GES-19 and GES-20 genes in strain PAO1 remains to be determined.

Class B β-lactamases increased the MIC of all β-lactams tested except for the monobactam aztreonam (Tables 2 and 4). These findings are consistent with previous findings in *E. coli* (32, 34, 37), as well as the known lack of activity of class B β-lactamases against monobactams (32), and suggest that aztreonam could be used to treat clinical isolates with only class B β-lactamases. The activity of Class D β-lactamases was largely limited to penicillins and cephalosporins, except for OXA-226 that increased the MICs for ceftazidime and aztreonam MICs. For example, OXA-10 increased piperacillin, ticarcillin, cefepime, and aztreonam MIC when expressed in *P. aeruginosa* PAO1 (Table 2), consistent with an earlier study (85).

The β-lactamase inhibitor tested here, tazobactam, is commonly co-administered with β-lactams and has been reported to inhibit many class A and some class C and D β-lactamases, although not KPC β-lactamases (47, 73, 86–88). Tazobactam inhibits GES-2 that has a very similar sequence (two or three amino acids different) to both GES-19 and GES-20 (88) but did not affect the MIC of *P. aeruginosa* PAO1 expressing GES-19 or GES-20 (Table 5) indicating that the differences between enzymes are sufficient to make GES-19 and GES-20 resistant to tazobactam. Instead, tazobactam only showed activity against the class D β-lactamases OXA-21 and OXA-226 (Table 5). Tazobactam also has activity against OXA-2 (47) that is very similar to OXA-226 and relatively closely related to OXA-21. These findings indicate that tazobactam likely has activity against other class D β-lactamases closely related to OXA-2 but has variable effects on class A enzymes. Other β-lactamase inhibitors, such as avibactam and relebactam, are effective against class A β-lactamases (89, 90). It will be of importance to investigate the activities of a range of β-lactamase inhibitors against enzymes expressed in strain PAO1 to enable comparison with direct enzymatic assays (86, 91). Relating this information to the frequencies at which different β-lactamases are present in *P. aeruginosa* may allow fine-tuning of antibiotic/inhibitor combinations in treating *P. aeruginosa* infections.

Although acquired *bla* genes are at a low but significant level (3%) in the *P. aeruginosa* population, they are enriched in carbapenem-resistant isolates (35% of isolates), indicating their importance to resistance. It should be noted that most currently available bacterial genome sequences, including almost all of those analyxed here, are assembled from short read-data that leads to poor assembly of mobile elements (92, 93), and so the frequency of *bla* genes could be significantly underestimated in both carbapenem-resistant and carbapenem-susceptible sub-populations of *P. aeruginosa*. Knowing the distribution and frequency of acquired β-lactamases in the *P. aeruginosa* population, and in other bacteria, can contribute to development of approaches to manage the spread of the threat.

Most clinical isolates harboring *bla* genes had higher MICs than *P. aeruginosa* PAO1 expressing individual genes (Tables 2 and 3). For example, unlike strain PAO1 all five isolates with a class A β-lactamase were clinically resistant towards all β-lactams (Table 3),

except that one isolate had only intermediate resistance towards piperacillin. The higher MICs relative to strain PAO1 expressing individual *bla* genes could be due in part to the presence of multiple *bla* genes in many of the isolates. About a quarter (24.5%) of isolates that have class B β-lactamases also have either a class A or D β-lactamase, and over half (54%) of isolates with a class D enzyme also have other acquired β-lactamases (Table 3; Tables S1 and S2). However, the higher MICs of the clinical isolates could also be due to the presence of mutations affecting antibiotic efflux or uptake, or mutations in penicillin-binding proteins. For example, mutations inactivating OprD explain the finding that deletion of *bla* genes encoding enzymes with carbapenemase activity (GES-20, VIM-2 and IMP-54) reduced MICs of carbapenems and other β-lactams, it did not render the bacteria carbapenem-susceptible (Table 4). The relative sensitivity of strain PAO1 to β-lactams means that MICs of this strain expressing individual β-lactamases can likely be used to indicate the minimum MICs of different antibiotics for clinical isolates with the same β-lactamases. *bla*-containing isolates and carbapenem-resistant isolates encompass a large range of the genetic diversity of the *P. aeruginosa* species. This indicates that *bla* genes have been acquired by *P. aeruginosa* on multiple occasions, and that most if not all *P. aeruginosa* isolates are capable of acquiring *bla* genes and becoming carbapenem-resistant. Notably, high-risk clones, such as ST235, that are associated with poorer patient outcomes following infection (61–63) have acquired a range of different *bla* genes. The genes encoding VIM-2 and OXA-10 are co-located on a transposon in the chromosome, as well as those encoding GES-19, GES-20, and OXA-2. Almost identical transposons were identified in a wide range of Gram-negative genera, indicating that the transposons have likely been acquired by *P. aeruginosa* from other species. The presence of *bla* genes on mobile genetic elements that frequently contain multiple resistance genes provides opportunities to acquire multiple *bla* genes simultaneously. A significant proportion of *bla*-containing isolates (28%) had multiple *bla* genes. The GES-19 and GES-20 genes are adjacent to one another in all three clinical isolates that contain them (Table 3; Table S2; Fig. S1b) and are often reported together in literature (94, 95). The presence of multiple different *bla* genes on the same element indicates that combinations are selected most likely due to additive or synergistic effects of different enzymes in conferring resistance to a wider range of antibiotics. The transposons also carry genes for resistance to aminoglycoside antibiotics, emphasizing how multiple antibiotic resistance genes can be acquired in a single genetic event.

The *bla*KPC-2 gene is located on a plasmid, pPAE1255885 (Fig. S2). Strain PAO1 containing this plasmid had very similar MIC values to the clinical isolate containing the same plasmid (Table 6), indicating that the plasmid may be responsible for the majority of β-lactam resistance in this isolate. The plasmid resulted in significantly higher MICs for strain PAO1 than KPC-2 expressed alone (Tables 2 and 6), although KPC-2 is the only gene on pPAE1255885 known to contribute directly to antibiotic resistance. The difference may be due to increased expression of KPC-2 from the multi-copy plasmid or may indicate the presence of an uncharacterized mechanism of β-lactam resistance on pPAE1255885. Collectively, these examples emphasize how mobile elements can carry a diverse range of β-lactamases and other antibiotic resistance genes. Many mobile elements are poorly characterized, and their prevalence in bacterial populations is not well understood. However, understanding and reducing the spread of mobile elements will likely be important in combating antibiotic resistance.

In conclusion, acquired β-lactamases contribute towards resistance to a large range of β-lactams in *P. aeruginosa*. Different β-lactamases of both the same and different classes have differences in their target β-lactams and also the extent of their effect on MIC. β-lactamases likely play a significant role in carbapenem resistance as they are highly enriched in carbapenem-resistant *P. aeruginosa* clinical isolates. Because many of the β-lactamases studied here are present in a wide range of bacterial species (54), determining their activities has potential to inform decisions on antibiotic treatment of infections by other bacterial species nd *P. aeruginosa*.

## MATERIALS AND METHODS

### Bacteria and growth conditions

Bacterial strains and plasmids used in this study are listed in Table S9. Bacteria were grown at 37 °C in Lysogeny broth (LB) with shaking (200 rpm) or on Luria agar (LA) unless stated otherwise, with additional chemicals added as required. Gentamicin (*E. coli*, 20 µg/mL; *P. aeruginosa*, 30 mg/L) was used to select for bacteria containing plasmid pJN105 tetracycline (*E. coli*, 12.5 mg/L; *P. aeruginosa*, 25 mg/L) for bacteria containing plasmid pEX18Tc or pSW196 and meropenem (2 mg/L) for *P. aeruginosa* strain PAO1 containing plasmid pPAE1255885. For conjugation, ST18 was used as a donor strain and grown in media supplemented with δ-aminolaevulinic acid (ALA) (50 mg/L), and *P. aeruginosa* recipients were grown at 42°C in LB with 0.4% $KNO_3$ as described previously (96–98). For expression of cloned genes from plasmid pSW196, arabinose (5 mg/mL) was included in the growth medium.

### Molecular biology methods

Genomic DNA was purified using the UltraClean Microbial Kit (Qiagen, Hilden, Germany), plasmids were purified using the NucleoSpin Plasmid kit (Macherey Nagel, Dueren, Germany), and PCR products were purified and extracted from agarose gels using the NucleoSpin Gel and PCR Clean-up kit (Macherey Nagel, Dueren, Germany) according to the manufacturers' instructions. For DNA cloning, PCR amplification was carried out with genomic DNA from isolates listed in Table S9 as template and with appropriate primers (Table S10). Amplicons for expression of the *bla* genes included the predicted ribosome-binding sites. Amplicons were cloned into pSW196 (99) for expression of cloned genes in *P. aeruginosa*. DNA cloning was carried out using restriction enzymes and T4 DNA ligase (New England Biolabs, Ipswich, MA, USA). DNA sequencing (Genetics Analysis Service, University of Otago, Dunedin, New Zealand) was carried out with appropriate primers on all plasmid constructs to confirm that the intended plasmids had been obtained with no mutations.

### Genetic manipulation

*bla* genes that had been cloned into pSW196 were transferred into the chromosome of the β-lactam-susceptible reference strain PAO1 by conjugation (97). Plasmid pPAE1255885 was transformed into *P. aeruginosa* PAO1 that had been made competent by the sucrose washing method (100). *bla* and *ampC* genes were deleted from the genomes of *P. aeruginosa* isolates using two-step allelic exchange as previously described (96–98). In brief, PCR amplification with appropriate primers (Table S10) was used to amplify ~1-kb fragments flanking the gene(s) to be deleted, using genomic DNA from the relevant isolate as template. The PCR fragments were cloned adjacently into pEX18Tc (101). Plasmid pEX18Tc containing the two cloned fragments was transferred into the chromosome of recipient strains by conjugation from *E. coli* ST18, followed by homologous recombination. Mutants in which the engineered deletion alleles had replaced the wild-type DNA were identified by sucrose selection, followed by PCR amplification and sequencing using primers diagnostic for the presence of the deletion.

### Minimum inhibitory concentration (MIC) testing

Overnight cultures of *P. aeruginosa* were diluted to $1.5 \times 10^6$ CFU/mL and two 5-µL aliquots were spotted onto LA plates supplemented with doubling concentrations of antibiotic (102, 103), and, if required, arabinose to induce expression of cloned genes. Tazobactam was used at a set concentration of 4 µg/mL. Agar plates were incubated for 24 h, and the antibiotic MIC was defined as the lowest concentration that inhibited growth. CLSI breakpoints were utilized to classify resistant and susceptible phenotypes (68). Meropenem and imipenem have resistance breakpoints of 8 mg/L, aztreonam,

ceftazidime, and cefepime of 32 mg/L, piperacillin of 64 mg/L, and ticarcillin of 128 mg/L. CLSI does not have thresholds for carbenicillin, which is not used clinically.

## Bioinformatic analysis

Genome sequences were determined by the International *Pseudomonas* Consortium Database (IPCD) (56, 57), by Khaledi et al. (58), and for isolates in our laboratory collection (96). Genome sequences were from isolates from five continents and included isolates from a range of human infections and from environmental and animal sources (Tables S1 to S3). ResFinder V 4.0 was used to identify acquired antibacterial resistance genes from draft genome sequences (104). The identified acquired resistance genes were verified using the CARD BLAST (https://card.mcmaster.ca/) (105). For genomes with complete fully assembled sequences, mobile genetic elements, the genes present on them and the boundaries of transposons were identified with TnCentral BLAST (106) and NCBI BLAST (107). For phylogenetic analysis, ParSNP version 1.2 from the Harvest suite 1.1.2 was used to construct a core genome alignment of isolates with *P. aeruginosa* PAO1 as the reference strain (108, 109). The phylogenetic trees were visualized on TreeGraph 2 (110). The MLST type was determined using FastMLST (https://github.com/EnzoAndree/FastMLST) (111).

## ACKNOWLEDGMENTS

We very gratefully acknowledge Roger Levesque, Bob Hancock, Craig Winstanley, and co-workers for generously providing isolates used in this study and for generating many of the genome sequences.

K.A.G. was supported by a postgraduate scholarship from the University of Otago. Additional research funding was provided by the University of Otago. The funder had no role in study design, data collection and interpretation, or the decision to submit the work for publication.

K.A.G.: Project design, investigation, data collection, data analysis, writing the original draft, reviewing, and editing of the manuscript. I.L.L.: Project design, project administration, supervision, reviewing, and editing of the manuscript.

Both authors read and agreed to the submitted version of the manuscript.

## AUTHOR AFFILIATION

[1]Department of Biochemistry, University of Otago, Dunedin, New Zealand

## AUTHOR ORCIDs

Karl A. Glen ⓘ http://orcid.org/0000-0002-9370-3893
Iain L. Lamont ⓘ http://orcid.org/0000-0002-7601-6354

## FUNDING

| Funder | Grant(s) | Author(s) |
| --- | --- | --- |
| University of Otago (Te Whare Wānanga o Otāgo) | | Iain L. Lamont |

## AUTHOR CONTRIBUTIONS

Karl A. Glen, Conceptualization, Formal analysis, Funding acquisition, Investigation, Validation, Visualization, Writing – original draft, Writing – review and editing | Iain L. Lamont, Conceptualization, Funding acquisition, Project administration, Supervision, Writing – original draft, Writing – review and editing

## ADDITIONAL FILES

The following material is available online.

## Supplemental Material

**Figures S1 and S2 (Spectrum00694-24-s0001.pdf).** Figure S1: Extended phylogenetic tree. Figure S2: Maps of acquired genetic elements.
**Tables S1 to S10 (Spectrum00694-24-s0002.xlsx).** Supplemental tables detailing genomes analyzed, MIC replicate data, and methods-related materials.

## Open Peer Review

**PEER REVIEW HISTORY (review-history.pdf).** An accounting of the reviewer comments and feedback.

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
