## [Reviewer comments · Microbiology Spectrum]

Microbiology Spectrum

Characterization of acquired β -lactamases in *Pseudomonas aeruginosa* and quantification of their contributions to resistance

Karl Glen and Iain Lamont

Corresponding Author(s): Iain Lamont, University of Otago

Review Timeline:

Submission Date:	March 15, 2024
Editorial Decision:	May 20, 2024
Revision Received:	June 14, 2024
Editorial Decision:	July 8, 2024
Revision Received:	July 23, 2024
Accepted:	July 25, 2024

Editor: Pablo Power

Reviewer(s): Disclosure of reviewer identity is with reference to reviewer comments included in decision letter(s). The following individuals involved in review of your submission have agreed to reveal their identity: Daniela Cejas (Reviewer #2)

Transaction Report:

DOI: <https://doi.org/10.1128/spectrum.00694-24>

Re: Spectrum00694-24 (Characterization of acquired β -lactamases in *Pseudomonas aeruginosa* and quantification of their contributions to resistance)

Dear Prof. Iain L Lamont:

Thank you for the privilege of reviewing your work. Below you will find my comments, instructions from the Spectrum editorial office, and the reviewer comments.

Revision Guidelines

Sincerely,
Pablo Power
Editor
Microbiology Spectrum

Reviewer #1 (Comments for the Author):

This work assesses the prevalence and impact on resistance of acquired β -lactamases from *Pseudomonas aeruginosa*. While overall some useful information is provided, there are also several points for the authors to address:

1. Lines 71-73. Might be informative to mention novel β -lactams and β -lactamase inhibitor combinations here. Moreover, the

study could be improved by providing data on their activity against β -lactamase producing strains.

2. Lines 111-113. How representative is this world-wide collection? It is a relevant aspect for the implications of this survey and therefore it should be explained in detail.

3. Same applies to the meropenem resistant collection (lines 119-124).

4. Line 140 "Substrate specificity of β -lactamases" Most of the described data on this and the following subheading is already well known.

5. Line 172 "Deletion of carbapenemase genes from clinical isolates" The information of this subheading is useful, but it would be desirable to determine which underlying resistance mechanisms are behind carbapenem resistance besides carbapenemase production.

6. Line 199 "Effectiveness of tazobactam on *P. aeruginosa* containing acquired β -lactamase genes" Would be useful to compare with avibactam.

Reviewer #2 (Comments for the Author):

The authors analyzed selected 858 genomes of *P. aeruginosa* with unknown antibiotic phenotype looking for the presence of bla.

232 out of 858 genomes harbored different bla genes. Then, the phylogenetic relationship among the isolates was studied. In addition, representative bla belonging different Ambler class were cloned in PAO-1 and the susceptibility of different beta-lactams was determined.

Then, the deletion of blaVIM-2, blaGES-20 and blaIMP-54 was performed, and the MIC to beta-lactams were evaluated in the mutants. These bla alleles were selected because they increased the MIC to meropenem in PAO-1.

In general, the manuscript presents so much work and results, but its writing should be revised highlighting the importance of the results and clarifying the inclusion criteria of the genomes and isolates. In addition, the manuscript has 1 figure and 6 tables but also the authors added 10 tables and 2 figures in supplementary material, making it difficult to read. The author must find a simple way to present their results.

Comments by section:

Objective: Add which collection of isolates was subjected to study.

Material & Method:

Include a summary about the isolates that were analyzed in this study (your own and from database). Explain your selection criteria, mentioning how did you choose your isolates and worldwide genomes.

In general, what was the origin of the isolates? (clinical isolates recovered from inpatients, animal, environmental?)

Regarding the phylogenetic analysis: The authors analyze the relationship among 137 isolates, where 110 harbored different bla. The assignment of the sequence type (ST) of each isolate is a prior step before to analyze their phylogenetic relationship, to understand the importance of the resistance markers in the group of the isolates.

Results:

Line 116 (add in text which class D enzyme was most frequent).

Why did you only include meropenem from carbapenemes as inclusion criteria to investigate bla_{MBL}? (line 121)

Line 122-124, mention which acquired bla was the most prevalent in the 61 isolates, and 23 isolates.

Line 132-138: Perhaps determining the ST you can assign which are the successful clones that you mentioned in line 137. I couldn't find the PA7 subgroup in figure 1.

Line 140-170: Determine the susceptibility of some of the new line of antibiotics that are available for the serine-carbapenemase producers such as ceftazidime/avibactam, imipenem/relebactam or ceftolozane/tazobactam.

Discussion:

Line 274-276: Modify the sentence considering that aztreonam is a monobactam, and class B beta-lactamases are not active against it.

Put Enterobacteriaceae in italic (line 100 and line 222).

Microbiology Spectrum 00694-24

Point-by-point response to reviewers' comments.

Reviewers' comments are in normal font. Our responses are in red.

Reviewer #1 (Comments for the Author):

This work assesses the prevalence and impact on resistance of acquired β -lactamases from *Pseudomonas aeruginosa*. While overall some useful information is provided, there are also several points for the authors to address:

1. Lines 71-73. Might be informative to mention novel β -lactams and β -lactamase inhibitor combinations here. Moreover, the study could be improved by providing data on their activity against β -lactamase producing strains.

We have now included information on novel b-lactams (line 73-75) and expanded the existing text on inhibitors (line 95). We agree that a comprehensive study on the effects of new antibiotics and b-lactamase inhibitors on b-lactamase-producing bacteria (both clinical isolates, and a reference strain engineered to express different b-lactamases) would be informative in assessing the potential uses of new compounds. However, such a study is beyond the scope of this manuscript which is focussed on antibiotics in current clinical use.

2. Lines 111-113. How representative is this world-wide collection? It is a relevant aspect for the implications of this survey and therefore it should be explained in detail.

We have now expanded the text (lines 116-120) to provide references and more fully describe the collection used for this analysis. We have also described the sources of the sequenced isolates in the materials and methods (lines 426-430).

The sources of all isolated identified as containing b-lactamases (Figure S1) are listed in Tables S1, S2 and S3 as indicated in the figure legends. The sources of isolates for which carbapenem phenotypes had been determined (Figure 1) are listed in Table S3, as indicated in the figure legend. Isolates for this figure were sourced from North America; Central America; South America; Europe; Asia; and Australasia. Isolates for Figure S1 were predominantly from a variety of hospitals in Europe although some isolates from other continents were also included. We have expanded the text (line 129-130) to emphasise the diversity of the isolates that were used.

3. Same applies to the meropenem resistant collection (lines 119-124).

We have expanded the text (lines 129-130) to explain the sources of these isolates. Full information is included in Table S2.

4. Line 140 "Substrate specificity of β -lactamases" Most of the described data on this and the following subheading is already well know.

Information on substrate antibiotics of b-lactamases is summarised in the introduction (lines 104-106), making the point that the substrates of many b-lactamases are not yet known.

This is also indicated in the relevant results section (lines 157-159; we have slightly modified the wording to better emphasise this point). We have repeated our literature search to identify previous research on the substrate specificities of the enzymes characterised in our study. So far as we can determine, the substrates for GES-20, IMP-54, OXA-21, OXA-129 and OXA-226 have not been identified previously. This is also mentioned in the Discussion (lines 267-269). The substrates of KPC-2, GES-19 and VIM-2 have been investigated previously for other species but their impacts on antibiotic susceptibility of *P. aeruginosa* have not been previously investigated.

5. Line 172 "Deletion of carbapenemase genes from clinical isolates" The information of this subheading is useful, but it would be desirable to determine which underlying resistance mechanisms are behind carbapenem resistance besides carbapenemase production.

We have analysed the genomes of the relevant isolates. All have mutations that inactivate OprD, that is strongly associated with carbapenem susceptibility, explaining their resistance phenotypes. We have added this information to the text (lines 205-207 and also line 330).

6. Line 199 "Effectiveness of tazobactam on *P. aeruginosa* containing acquired β -lactamase genes" Would be useful to compare with avibactam.

As well as identifying which enzymes are targeted by tazobactam, the results with tazobactam in this section indicate the utility of our approach. We agree that comparison with other β -lactamase inhibitors, including avibactam, would be of interest. However, an extended analysis of the effectiveness of a range of inhibitors against multiple enzymes is beyond the scope of this study.

Reviewer #2 (Comments for the Author):

The authors analyzed selected 858 genomes of *P. aeruginosa* with unknown antibiotic phenotype looking for the presence of bla.

232 out of 858 genomes harbored different bla genes. Then, the phylogenetic relationship among the isolates was studied.

In addition, representative bla belonging different Ambler class were cloned in PAO-1 and the susceptibility of different beta-lactams was determined.

Then, the deletion of blaVIM-2, blaGES-20 and blaIMP-54 was performed, and the MIC to beta-lactams were evaluated in the mutants. These bla alleles were selected because they increased the MIC to meropenem in PAO-1.

In general, the manuscript present so much work and results, but its writing should be revised highlighting the importance of the results and clarifying the inclusion criteria of the genomes and isolates. In addition, the manuscript has 1 figure and 6 tables but also the authors added 10 tables and 2 figures in supplementary material, making it difficult to read. The author must find a simple way to present their results.

The reviewer makes a number of important points. The importance of the results is described in the "Importance" section. We have expanded this section slightly to further emphasise the importance of the results. We also indicate the importance of the results in the final paragraph of the discussion, and have added an additional sentence to the discussion (lines 262-264).

The number of figures and tables is similar to other papers published recently in Microbiology Spectrum eg. Martiny et al Microbiology Spectrum 04108-23 (4 figures, 1 table; supplementary material 1 table, 8 figures) or Liu et al Microbiology Spectrum 00450-24 (6 figures; 3 tables and 7 figures in the supplementary material). We considered the possibility of merging tables in the main text but each table deals with a separate topic and merging them would not increase clarity. The supplementary material provides information that is not essential for understanding of the main text, but provides important and necessary supporting detail for the interested reader and also raw data that is summarised in tables in the main text.

We have now modified the text to more fully describe the sources of isolates used in this study, as described in the response to reviewer one.

Comments by section:

Objective: Add which collection of isolates was subjected to study.

We have modified the text to more fully describe the sources of isolates used in this study, as described above. Collections used in the study, including their origins, are listed in Supplementary Tables S1-S3.

Material & Method:

Include a summary about the isolates that were analyzed in this study (your own and from database). Explain your selection criteria, mentioning how did you choose your isolates and worldwide genomes.

In general, what was the origin of the isolates? (clinical isolates recovered from in patients, animal, environmental?)

We have now modified the text to more fully describe the sources of isolates used in this study, as described in the response to reviewer one. As different data sets were used for different analyses, we have described each dataset in the results rather than in the Materials and Methods.

Regarding the phylogenetic analysis: The authors analyze the relationship among 137 isolates, where 110 harbored different bla. The assignment of the sequence type (ST) of each isolate is a prior step before to analyze their phylogenetic relationship, to understand the importance of the resistance markers in the group of the isolates.

We have now carried out MLST analysis, as recommended by the reviewer – this was a very good suggestion. MLST types have been added to Figures 1 and S1 and align with the phylogeny determined by whole genome analysis. We have added comment on the MLST typing in the text (lines 147-150 in results; and also one sentence in discussion (lines 340-342))

Results:

Line 116 (add in text which class D enzyme was most frequent).

We have now included this information for each of classes D, A and B (lines 122-123 in the revised manuscript).

Why did you only include meropenem from carbapenemes as inclusion criteria to investigate bla_{MBL}? (line 121)

For the majority of the isolates, meropenem was the only carbapenem for which MIC data had been determined (Khaledi et al., reference 58 in the manuscript). We are not aware of large-scale studies on MIC data for other carbapenems on genome-sequenced isolates.

Line 122-124, mention which acquired bla was the most prevalent in the 61 isolates, and 23 isolates.

We have now included this information (lines 132-133 in the revised manuscript).

Line 132-138: Perhaps determining the ST you can assign which are the successful clones that you mentioned in line 137. I couldn't find the PA7 subgroup in figure 1.

We have now carried out MLST typing, as described above. We have also added comment on the most prevalent ST types (lines 147-150). Isolates in the PA7 subgroup were not present in the datasets used for Figure 1, as noted in the manuscript (lines 143-144). This reflects a lack of *bla* genes in this small subgroup.

Line 140-170: Determine the susceptibility of some of the new line of antibiotics that are available for the serine-carbapenemases producers such as ceftazidime/avibactam, imipenem/relebactam or ceftolozane/tazobactam.

We agree that it will be of interest to carry out an extensive study on the effectiveness of antibiotic-inhibitor combinations on MIC, and the effects of inhibitors on β -lactamases. However, as noted in our response to reviewer 1, such a study is beyond the scope of this research. Here we have focussed on the effects of the β -lactams, with inclusion of tazobactam to demonstrating that our approach will also be valid for investigating antibiotic/inhibitor combinations. We are also sensitive to the comment by the reviewer that our manuscript already contains a large amount of data.

Discussion:

Line 274-276: Modify the sentence considering that aztreonam is a monobactam, and class B beta-lactamases are not active against it.

We have reworded this part of the text (lines 286-288 in the revised manuscript).

Put Enterobacteriaceae in italic (line 100 and line 222).

We have made these changes.

Re: Spectrum00694-24R1 (Characterization of acquired β -lactamases in *Pseudomonas aeruginosa* and quantification of their contributions to resistance)

Dear Prof. Iain L Lamont:

Thank you for the privilege of reviewing your work. Below you will find my comments, instructions from the Spectrum editorial office, and the reviewer comments.

Both reviewers agreed that the manuscript has been improved and that most of their concerns and suggestions have been satisfactorily addressed. However, a few additional modifications are requested by one of them. Please return the manuscript within 60 days; if you cannot complete the modification within this time period, please contact me. If you do not wish to modify the manuscript and prefer to submit it to another journal, notify me immediately so that the manuscript may be formally withdrawn from consideration by Spectrum.

Revision Guidelines

Sincerely,
Pablo Power
Editor
Microbiology Spectrum

Reviewer #1 (Comments for the Author):

All points raised to the previous version have been satisfactorily addressed and therefore I have no further comments for the authors consideration

Reviewer #2 (Comments for the Author):

The article is better than its original version but its writing could be still improved to make it more easy to read.

* Since 2024, carbapenem resistant *Pseudomonas aeruginosa* is not anymore a critical pathogen accordingly WHO. Now it is belong to the high group of pathogens. Actualize this in the introduction and reference 16.

* Replace SENSITIVE by SUSCEPTIBLE in the manuscript when you refer to the inability to grow in the presence of the antibiotics.

* Line 88, take into consideration that only serine-carbapenemase (e.g. KPC) can hidrolize all beta-lactams.

* Re-write paragraph 104-108 considering that is a big amount of information about beta-lactamases in *P. aeruginosa*. If you use the keywords *Pseudomonas aeruginosa* + beta-lactamases in pubmed: 3,847 results can be found.

* Line 122-124: display the percentage of isolates/genomes carrying OXA-10, CARB-2, and VIM-2 among class D, A and B beta-lactamases, respectively.

* Line 132-133: Following the sentence:"The most common β -lactamases found in carbapenem-resistant isolates for each class were GES-7 for Class A, VIM-2 for Class B and OXA-2 for Class D".... Briefly explain why the isolates carrying GES-7 and OXA-2 could be resistant to meropenem.

* Line 165: add which enzyme of class A and class B increased the MIC.

* As far as I know, blaVIM-2, blaGES-19 and blaGES-20 are usually encoded by plasmids. Detail how did you determined the genetic location of these genes.

* Line 258: you said Tazobactam had limited activity against the enzymes characterized in this study, which is correct, but I believe that it is necessary to add a discussion about the importance of avibactam and relebactam (even if you did not determined their activity in your work).

* Line 265: please revise if the effect of VIM-2 is not previously investigated in *P. aeruginosa*. This enzyme is the most prevalent meta-beta-lactamase in *P. aeruginosa*.

* Line 340: How did you realise that blaVIM-2 is located on the chromosome? Usually this marker is plasmid encoded.

The article is better than its original version but its writing could be still improved to make it more easy to read.

*Since 2024, carbapenem resistant *Pseudomonas aeruginosa* is not anymore a critical pathogen accordingly WHO. Now it is belong to the high group of pathogens. Actualize this in the introduction and reference 16.

* Replace SENSITIVE by SUSCEPTIBLE in the manuscript when you refer to the inability to grow in the presence of the antibiotics.

* Line 88, take into consideration that only serine-carbapenemase (e.g. KPC) can hidrolize all beta-lactams.

* Re-wrtite paragraph 104-108 considering that is a big amount of information about beta-lactamases in *P. aeruginosa*. If you use the keywords *Pseumonas aeruginosa* + beta-lactamases in pubmed: 3,847 results can be found.

* Line 122-124: display the percentage of isolates/genomes carrying OXA-10, CARB-2, and VIM-2 among class D, A and B beta-lactamases, respectively.

* Line 132-133: Following the sentence:"The most common β -lactamases found in carbapenem-resistant isolates for each class were GES-7 for Class A, VIM-2 for Class B and OXA-2 for Class D".... Briefly explain why the isolates carrying GES-7 and OXA-2 could be resistant to meropenem.

* Line 165: add which enzyme of class A and class B increased the MIC.

* As far as I know, *bla*_{VIM-2}, *bla*_{GES-19} and *bla*_{GES-20} are usually encoded by plamsids. Detail how did you determined the genetic location of these genes.

* Line 258: you said Tazobactam had limited activity against the enzymes characterized in this study, which is correct, but I believe that it is necessary to add a discussion about the importance of avibactam and relebactam (even if you did not determined their activity in your work).

* Line 265: please revise if the effect of VIM-2 is not previously investigated in *P. aeruginosa*. This enzyme is the most prevalent meta-beta-lactamase in *P. aeruginosa*.

* Line 340: How did you realise that *bla*_{VIM-2} is located on the chromosome? Usually this marker is plasmid encoded.

Microbiology Spectrum 00694-24

Point-by-point response to reviewers' comments.

Reviewers' comments are in normal font. Our responses are in red.

Reviewer #2 (Comments for the Author):

The article is better than its original version but its writing could be still improved to make it more easy to read.

*Since 2024, carbapenem resistant *Pseudomonas aeruginosa* is not anymore a critical pathogen accordingly WHO. Now it is belong to the high group of pathogens. Actualize this in the introduction and reference 16.

Thank you for pointing out this error. The text has been modified accordingly (lines 73-74 in the revised manuscript).

* Replace SENSITIVE by SUSCEPTIBLE in the manuscript when you refer to the inability to grow in the presence of the antibiotics.

Sensitive has been replaced with susceptible throughout the manuscript, as recommended by the reviewer and consistent with CLSI terminology – thank you for pointing this out.

* Line 88, take into consideration that only serine-carbapenemase (e.g. KPC) can hidrolize all beta-lactams.

The sentence has been adjusted (Line 91) to accommodate this point, noting that most enzymes are only active against a subset of beta-lactams

* Re-wrtite paragraph 104-108 considering that is a big amount of information about beta-lactamases in *P. aeruginosa*. If you use the keywords *Pseumonas aeruginosa* + beta-lactamases in pubmed: 3,847 results can be found.

We fully agree that there are very many papers describing the beta lactamases of *P. aeruginosa*, many on the intrinsic beta lactamases and others describing the presence of acquired beta lactamases. The focus of this paragraph is not the presence of beta lactamases or the activity of intrinsic enzymes, but the lack of information on the substrates of acquired enzymes and the effects of these enzymes on antibiotic susceptibility. A search of PubMed with “*Pseudomonas aeruginosa* beta lactamases substrates” gives less than 200 results, most of which do not relate to the knowledge gaps that we outline in this paragraph. We have expanded the paragraph (lines 109-112) to emphasise that while a relatively large number of studies have documented the presence of horizontally-acquired b-lactamases in *P. aeruginosa*, there is very little quantitative information on the effects of these enzymes on antibiotic susceptibility.

* Line 122-124: display the percentage of isolates/genomes carrying OXA-10, CARB-2, and VIM-2 among class D, A and B beta-lactamases, respectively.

We have now reworked this part of the text to incorporate these percentages (lines 125-128)

* Line 132-133: Following the sentence:"The most common β -lactamases found in carbapenem-resistant isolates for each class were GES-7 for Class A, VIM-2 for Class B and OXA-2 for Class D".... Briefly explain why the isolates carrying GES-7 and OXA-2 could be resistant to meropenem.

There was a typographical error in this sentence, "GES-7" should have read "GES-5". We apologise for this error that has misled the reviewer. GES-5 and OXA-2 are both active against meropenem (Stewart et al., *Biochemistry* 54:588-597 [2015]) (Arca-Suárez, et al., *Antimicrobial Agents and Chemotherapy* 63: e01110-19 [2019]). These activities are consistent with the presence of these enzymes in meropenem-resistant isolates. However the resistance/ susceptibility phenotype is also influenced by other genetic factors in the isolates and we have added a sentence to this effect (lines 138-141).

* Line 165: add which enzyme of class A and class B increased the MIC.

We have now added this information to the text (lines 174-175).

* As far as I know, blaVIM-2, blaGES-19 and blaGES-20 are usually encoded by plasmids. Detail how did you determined the genetic location of these genes.

The genomes of the isolates containing blaVIM-2, blaGES-19 and blaGES-20 genes were fully assembled using long-read sequencing, which showed that these genes were on mobile elements in the chromosome. Other studies have also reported that blaVIM-2 is commonly chromosomally-located in *P. aeruginosa* isolates that contain it (Vega et al, *Antimicrobial Agents and Chemotherapy* 68:e0147423); we have not identified prior studies on the genomic location of GES-19 and GES-20 genes in *P. aeruginosa*.

We described in the Results (lines 237-238) and in the Methods (line 455) that analysis of transposons (including those carrying these three genes) was only carried out for isolates where completed genome sequences (ie. generated with long-read sequencing) were available. Fully assembled genomes allow unambiguous assignment of the genes to a chromosomal location. We have expanded the text in both of these places to emphasise that the analysed genomes were fully assembled. As the genomes were sequenced and assembled prior to this study (as referenced in the paper) we have not repeated description of the sequencing/assembly process.

* Line 258: you said Tazobactam had limited activity against the enzymes characterized in this study, which is correct, but I believe that it is necessary to add a discussion about the importance of avibactam and relebactam (even if you did not determined their activity in your work).

Thank you for this suggestion, we have added a sentence about these inhibitors in the discussion (Lines 318-319).

* Line 265: please revise if the effect of VIM-2 is not previously investigated in *P. aeruginosa*. This enzyme is the most prevalent meta-beta-lactamase in *P. aeruginosa*.

We have reworded this sentence (lines 279-280) to emphasise that the effects of VIM-2 on a reference strain of *P. aeruginosa* have not been investigated previously. A number of studies have described the presence of VIM-2 in isolates of *P. aeruginosa*.

* Line 340: How did you realise that blaVIM-2 is located on the chromosome? Usually this marker is plasmid encoded.

The genomes of the isolates containing blaVIM-2 (and blaGES-19 and blaGES-20) were fully assembled using long-read sequencing, and the assemblies showed that all of these β -lactamase were on mobile elements in the chromosome, as described above.

Re: Spectrum00694-24R2 (Characterization of acquired β -lactamases in *Pseudomonas aeruginosa* and quantification of their contributions to resistance)

Dear Prof. Iain L Lamont:

Your manuscript has been accepted, and I am forwarding it to the ASM production staff for publication. Your paper will first be checked to make sure all elements meet the technical requirements. ASM staff will contact you if anything needs to be revised before copyediting and production can begin. Otherwise, you will be notified when your proofs are ready to be viewed.

Sincerely,
Pablo Power
Editor
Microbiology Spectrum